# LIonomers-New Generation of Ionomer: Understanding of Their Interaction and Structuration as a Function of the Tunability of Cation and Anion

**DOI:** 10.3390/polym15020370

**Published:** 2023-01-10

**Authors:** Liutong Hou, Sébastien Livi, Jean-François Gérard, Jannick Duchet-Rumeau

**Affiliations:** Université de Lyon, CNRS, Université Claude Bernard Lyon 1, INSA Lyon, Université Jean Monnet, UMR 5223, Ingénierie des Matériaux Polymères, CEDEX, F-69621 Villeurbanne, France

**Keywords:** LIonomers, ionomers, maleic anhydride grafted polypropylene, ionic liquids, ion and dipole interactions

## Abstract

In this work, by combining maleic anhydride-grafted polypropylene (PPgMA) and three different ionic liquids (ILs), i.e., tributyl (ethyl) phosphonium diethyl phosphate (denoted P^+^DEP), 1-ethyl-3-methylimidazolium diethyl phosphate (denoted EMIM DEP), and 1-ethyl-3-methylimidazolium acetate (denoted EMIM Ac), new ionic PP/IL polymer materials are generated and denoted as LIonomers. The structuration of ILs in LIonomers occurs from a nano/microphase separation process proved by TEM. NMR analyses reveal the existence of ionic–ionic and ionic–dipolar interactions between PPgMA and ILs within LIonomers. The rheological behavior of such IL/polymer combinations interpret the existence of interactions between maleic anhydride group and cation or anion composing the ionic liquid. These interactions can be tuned by the nature of cation (P^+^DEP vs. EMIM DEP) and anion (EMIM DEP vs. EMIM Ac) but also depend on the IL content. Thermal analyses demonstrate that IL could affect the crystallization process according to different pathways. Thanks to the maleic anhydride/IL interactions, an excellent compromise between stiffness and stretchability is obtained paving the way for processing new polyolefin-based materials.

## 1. Introduction

In the past decades, large attention has been poured into the study of ionomers. Ionomers are polymers containing up to 15 molar% of ionic groups that are frequently carboxylic, phosphoric, or sulfonic acid anions neutralized by metal salts, i.e., zinc, sodium or potassium cations [1,2]. These materials display a multi-scale structuration: the ion-pairs formed between acids anions and metal salts, usually assemble to form multiple states and participate in ionic clusters. Such ionic structuration leads to a restricted motion of polymer chains [3,4]. Consequently, a significant improvement of molten and solid-state properties are realized thanks to the physical crosslinking points generated via ionic interactions [5,6,7,8,9,10,11]. This effect is strongly related to the ionic multiplets generated as a segregated phase as for polyurethanes [12]. The rheological behavior (as for sulfonated polystyrene [10]) as well as the final crystalline state could be strongly affected by the existence of ion-pairs interactions. This effect is also demonstrated for the glass transition and melt elasticity of PLA-based ionomers [13,14]. According to the same approach, ionomers have been processed from different polymers such as polybutadiene-co acrylic acid or ethylene-methacrylic acid copolymers neutralized by zinc cations. Using maleic anhydride-grafted polypropylene (PPgMA) as precursor to design ionomers neutralized by zinc acetate leads to an increase of the molten strength due to the generation of branched chain structures [15,16]. Ionomer-like materials could be also used as efficient compatibilizers in non-miscible PP/PE blends [17,18]. For maleic anhydride-grafted polymers, zinc acetate and sodium hydrogen carbonate are usually considered. In fact, carboxylic functions generated from the hydration of the anhydride are known to interact with the corresponding cations. The establishment of interactions between grafted polyolefin chains and salts can be successfully realized via twin-screw extrusion. Such modification is demonstrated to be efficient for improving the elongation at break and impact properties. On theother hand, owing to the advantages of Ionic Liquids (ILs) such as thermal stability, non-flammability, and low saturated vapour pressure, they have been used as surfactants [19], plasticizers [20], compatibilizers [21], and structuring agents [22]. For example, 1-butyl-3-methyl imidazolium chloride could act as plasticizer and compatibilizer of starch/zinc blends, leading to smaller zinc aggregates [23]. Trihexyltetradecylphosphonium bis(trifluoromethylsulfonyl) imide and trihexyltetradecylphosphonium bis 2,4,4-(trimethylpentyl) phosphinate could serve as compatibilizers for thermoplastic blends (PP/PA6) [21]. In this study, ILs located at PP/PA interface behave as interfacial agents, resulting in the depressed size of dispersed phase and a decrease of the PP/PA interfacial tension. In fact, ionic liquids are promising components according to their intrinsic features and their ionic characters [19,24,25,26] which could promote interactions with polar groups.

Thus, a new generation of polypropylene-based ionomer is proposed for the first time in this study. New types of ionomers designed from Ionic Liquids (ILs) and maleic anhydride-grafted polypropylene (denoted as LIonomers) have been considered. An ionic interaction continuum must be generated from the interactions COO^−^ from PPgMA with the cation and the anion pairs of ILs. The microstructure shown in Figure 1 describes the targeted assembling of ionic liquid pairs leading to a percolated ionic species network within the grafted polyolefin matrix. This novel approach offers the way to tune the final LIonomers behavior by using different types of ILs. In this work, a phosphonium-based ionic liquid (PhIL), i.e., tributyl (ethyl)phosphonium diethyl phosphate (P^+^DEP, owning small steric hindrance favouring the interaction with polar groups of polymer matrix) and two imidazolium-based ionic liquids (ImILs), i.e., 1-ethyl-3-methylimidazolium diethyl phosphate (EMIM DEP) and 1-ethyl-3-methylimidazolium acetate (EMIM Ac), have been considered to investigate the role of cation, phosphonium vs. imidazoium and the role of anion, acetate vs. diethyl phosphate. This paper proposes to explore how (i) ILs generate ionic–ionic/dipole interactions with PPgMA, (ii) ionic–ionic/dipole interactions influence the PPgMA crystallization process and morphologies as well as consequently mechanical properties.

## 2. Experimental Section

### 2.1. Materials

OREVAC^®^ CA 100 from Arkema was considered as PPgMA. Ionic liquids supplied from TCI (Figure 1) were selected according to the nature of the cation (combined with the same anion), i.e., tributyl (ethyl) phosphonium diethyl phosphate (P^+^DEP) vs. 1-ethyl-3-methylimidazolium diethyl phosphate (EMIM DEP). The nature of the anion was also considered by comparing the same cation, i.e., 1-ethyl-3-methylimidazolium acetate (EMIM Ac) vs. 1-ethyl-3-methylimidazolium diethyl phosphate (EMIM DEP).

### 2.2. Material Preparation and Characterizations

The PPgMA was dried at 70 °C for 12 h. The PPgMA/IL mixtures (2 and 10 wt%) were blended into a DSM micro extruder at 180 °C for 5 min and injected at 30 °C. Three different LIonomer types, i.e., depending on the nature of the (cation/anion) pair and of IL content were prepared (Table 1).

The morphology of samples was carried out by transmission electron microscope (TEM, Phillips CM 120) at an accelerating voltage of 120 kV. To characterize PPgMA and LIonomer, they are cut using ultramicrotomy firstly and collected on a carbon film-coated cooper grid.

The chemical interactions between PPgMA and ILs were characterized by NMR spectroscopy. ^1^H and ^31^P NMR spectra of ionic liquids and LIonomers processed on PPgMA8 (8 wt% MA) were recorded using a Bruker Avance III 400 MHz at 373 K. Thanks to the relatively good solubility as well as the chemical nature (without any interactions with samples) of chlorobenzene-D5 (C_6_D_5_Cl), it has been selected as the NMR solvent. The chemical shifts (δ) are expressed in ppm relative to the internal reference C_6_D_5_Cl for ^1^H nuclei.

The rheological and dynamic mechanical analysis (DMA) were performed using a ARES-G2 rheometer (ARES-G2 SN#4010-0255, TA Instrument, New Castle, DE, USA). The rheological behavior was recorded at 180 °C using parallel plate geometry (d = 25 mm) under N_2_ atmosphere. Dynamic mechanical curves were recorded in the −80/100 °C temperature range with a heating rate of 3 K∙min^−1^.

Melting and crystallization curves were recorded by Differential Scanning Calorimetry (DSC, Q10 TA Instrument, New Castle, DE, USA) after erasing the thermal history under 5, 10, and 20 K∙min^−1^ heating and cooling rates. The crystallinity, $X_{c}$, was defined using Equation (1):(1)$$X_{c}=\frac{\Delta H_{f}}{\left(1-m\right)\Delta H_{f}{}^{0}}$$
where $\Delta H_{f}^{0}$ is 209 J/g, i.e., corresponding to the melting enthalpy of 100% crystallized polymer [27], $\Delta H_{f}$ being the melting enthalpy of the analyzed sample. Jeziorny’s [28] non-isothermal crystallization model was employed to analyze the crystallization process:(2)$$\ln\left[-\ln\left(1-X_{t}\right)\right]=\mathrm{nlnt}+\ln Z_{t}$$
(3)$$X_{T}=\frac{\int_{T_{0}}^{T_{c}}\left(\frac{dH_{c}}{dT}\right)/dT}{\int_{T_{0}}^{T_{\infty }}\left(\frac{dH_{c}}{dT}\right)/dT}$$
(4)$$t=\frac{T_{0}-T}{\varphi }$$
where $X_{t}$ is the relative crystallinity at crystallization time t; *n*, and $Z_{t}$ correspond to Avrami’ exponent and crystallization kinetic constant, respectively. Their values can be obtained by plotting $\ln\left[-\ln\left(1-X_{t}\right)\right]$ as a function of ln t. $X_{T}$ is related to the crystallinity with the change of temperature, $T_{0}$ and $T_{\infty }$ being the onset and the end of crystallization temperature; *T* is the temperature for a given crystallization time t, while φ is the cooling rate.

The crystallization half-time ($t_{1/2}$) could be determined from Equation (5):(5)$$t_{1/2}={\left(\frac{ln2}{Z_{t}}\right)}^{1/n}$$

The crystalline phase of the samples was analyzed by Wide Angle X-Ray Diffraction (WAXD, Bruker D8-Advance diffractometer) at room temperature (2θ = 10–45° using Cu Kα radiation − λ = 0.15406 nm). The diffractograms were further analyzed using Origin 2018 software considering Gauss’ fitting function. The crystal size was calculated according to the Scherrer’s equation:(6)$$D=\frac{K\lambda }{\left(\beta -s\right)cos\left(\theta \right)}$$
where D is the crystal size, *K* the crystalline shape parameter (≈0.89), β the measured sample diffraction peak half-height width, s the instrument factor calibration (0.05°), and θ the diffraction angle.

The crystalline morphology of PPgMA and LIonomers materials was observed by polarizing optical microscopy (POM). Materials were heated at 200 °C and remained for 3 min to eliminate the thermal history. Their crystal growth was observed after cooling at 120 °C with a cooling rate of 50 K∙min^−1^.

The strain–stress curves were recorded using an Instron tensile machine at room temperature operating at a strain speed of 50 mm∙min^−1^. Each sample (dumbbell-shaped specimen with the geometry T = 2 mm, w_0_ = 12 mm, w = 4 mm, L = 30 mm, D = 45 mm, L0 = 73.5 mm) has been repeated at least three times. In order to avoid the influence of water, all the prepared samples are stored in a dry and cold environment and placed in the oven at 80 °C for 4 h before test.

## 3. Results and Discussion

### 3.1. LIonomer Morphology

Transmission electron microscopy (TEM) is an efficient tool to study the phase separation of IL ion pairs within the polymer matrix (Figure 2). Owing to the expected morphologies of LIonomers, these granular “bean spots” could be attributed to phase-separated and associated ILs. IL-based aggregates can be also confirmed by EDX analyses as shown in Appendix A (spectra 1 in L-EMIM DEP-10: C: 85 wt%, N: 3 wt%, O: 11 wt%, P: 1 wt%; spectra 1 in polymer matrix in L-P^+^DEP-10: C: 97 wt%, O: 3 wt%; spectra 1 in L-P^+^DEP-10: C: 94 wt%, O: 5 wt%, P: 1 wt%; spectra 1 in L-EMIM Ac-10: C: 90 wt%, N: 2 wt%, O: 8 wt%). From comparison between L-P^+^DEP-10 (Figure 2c) and L-EMIM DEP-10 (Figure 2b), the previous one displays large ionic clusters, while the second one exhibits a multiscale structure based on not only large ionic liquid particles but also ionic multiplets (black spots) due to associated cation-anion pairs. These multiplets and particles dual microstructure are more pronounced in L-EMIM Ac-10 (Figure 2d), and numerous multiplets are observed compared to L-EMIM DEP-10 (Figure 2b). These various types of sub-structures of prepared LIonomers are strongly dependent on the ion pair nature of the ILs. In fact, it is possible to predict theoretically the aggregation of ion pairs or multiplet structures in a low dielectric medium. Both are influenced by polar–polar interactions that are responsible for the differences in size. The resulting morphology of the ionic liquid interactions is related to the equilibrium of interactions between the grafted-PP matrix and ILs. In fact, the size of the IL-rich phase could be tailored from nanoscale to microscale by a proper selection of the cation/anion combinations [29].

### 3.2. PPgMA–ILs Interactions

^1^H and ^31^P NMR spectroscopy were recorded to detect IL-polymer chain interactions from the comparison between neat IL spectra and the ones from corresponding LIonomer as shown in Figure 3 and Figure 4, respectively. In the case of EMIM Ac and the corresponding LIonomers, ^1^H NMR spectra reveal a shift of the protons on the cation (positions #6, #2, and #3 of EMIM Ac), from 10.85, 8.49, and 8.64 ppm [30,31] to higher fields, i.e., 10.09, 8.09, and 8.01 ppm. Such shifts could be attributed to the enhancement of electronic cloud density (e-cloud density) because of the ionic–ionic interactions between the cation and polymer matrix (Figure 5a). Thus, the COO^−^ group, i.e., the counter anion of the IL is positioned near the maleic anhydride group, then the counter anion participates in the opening of maleic anhydride leading to the formation of a new structure “ionic-branched chains” via the electrostatic interaction among cation/anion in EMIM Ac. In addition, the hydrogens in positions #2 and #3 of terminal cation may be positively shielded by C=O in the matrix. Besides, the hydrogen moving from 1.94 into 2.07 ppm at position #7 confirms the capture of the hydrogen carried by the imidazolium cation, thus inducing the generation of acetic anhydride group.

In addition to the enhanced e-cloud density at positions #2 and #3 (from 8.22 and 8.36 ppm of neat EMIM DEP [32,33] into 7.25 and 7.30 ppm of its LIonomer) as a result of the positive shielding effect of C=O in matrix (Figure 5b), the e-cloud density at position #6 decreases due to the strong electronic absorption capacity of oxygen in COO^−^ in matrix, thus its hydrogen shifts from 10.54 to 10.76 ppm. In fact, the same phenomenon is also observed with imidazolium IL combined with acetate or phosphate counter anions. In terms of the phosphonium IL combined with phosphate counter anion, the phosphorous compound of the anion displays a movement into low field (from 0.18 in EMIM DEP to 0.57 ppm in LIonomer) because of the effect of connected C=O group in matrix. In addition, the e-cloud density of phosphorus in P^+^DEP anion increases (^31^P NMR from −0.38 ppm into −0.95 ppm) since it lies in the positive shielded area induced by connected C=O (Figure 5c), while the strong electronegativity of oxygen in generated COO^−^ of matrix results in the migration in ^31^P NMR from 34.64 ppm into 34.99 ppm.

In summary, one may obtain that ***a*** all introduced ILs (EMIM Ac, EMIM DEP and P^+^DEP) can interact ionically with polar units in matrix; ***b*** the interaction structures between ILs and matrix varies with the change of anion/cation combinations; ***c*** the exceed ILs arrange as one ionic-branched chains using the generated COO^−^ group in PPgMA as the “head” and the cation/anion as the component via their original electrostatic interaction, such different interactions may connect the various dispersion of ILs in PPgMA and bring about some improvements into the overall performance (like molten viscosity, crystallization behavior, and so on).

### 3.3. Effect of ILs on Rheological Properties: Strength of Interactions

The molten state behavior of LIonomers has been greatly changed compared to the neat PPgMA one due to the existence of the ion–ion/ion–dipolar interactions described previously. All LIonomers display a strong shear thinning behavior compared to neat PPgMA as shown in Figure 6. This effect is more pronounced for L-EMMI Ac-10. Owing to the low viscosity of ILs (few mPa·s) [21], only the COO^−^/IL combined with IL/IL interactions could induce inter-chains connecting nanostructures as reported in ionomers [3]. Such ionic branching nanophases lead to a reduction of the chain mobility, resulting in a large increase of the melt viscosity [8,16,34,35,36]. For each LIonomers, the complex η⃰ value increases with the ILs content, as more ionic interchain connections are formed. Such viscosity values dependence with the nature of the ionic liquid and its content could be easily explained according to the previously reported interactions. In fact, the viscosity value of LIonomer processed with EMIM Ac is the highest one because the ionic interactions between imidazolium cation and carboxylic acid is the strongest one (three positions could create ionic–ionic interactions with carboxylic group as shown in Figure 5a). Additionally, the acetate anion is also highly polar and could be strongly associated with cation. These cation-anion pairs could also self-assemble to form multiplets and lead to this double scale structuration observed by TEM. For the LIonomer with 10 wt% EMIM DEP, a double structuration is observed with not only ionic liquid-rich particles but also ionic multiplets. Compared with LIonomers prepared with EMIM Ac, less fine phase separation is obtained due to ionic–ionic (1 position) and dipolar–ionic (2 positions) interactions (Figure 5b) generated to a lesser extent. The numerous multiplets act as branching agents, leading to increase the viscosity. The morphology as well as the strength of the interaction explain the ranking of viscosity values. Thus, clusters-based morphology, i.e., without larger multiplet structures, combined with the low intensity interactions established between the phosphonium cation and the carboxylate anion explain the lowest viscosity of LIonomers prepared with the phosphonium cation. Besides, the sequence of corresponding solid density, i.e., 0.909 g∙cm^−3^ of PPgMA < 0.910 g∙cm^−3^ of L-P^+^DEP-10 < 0.915 g∙cm^−3^ of L-EMIM DEP-10 < 0.917 g∙cm^−3^ of L-EMIM Ac-10 at around 20 °C could reveal the generation of ionic network within LIonomers, which could also reflect the interaction strength and is consistent with the resulted ranking of viscosity values.

### 3.4. Microstructure of LIonomers

Semi-crystalline polymers are considered to be a heterogeneous medium including crystalline fraction (CF), rigid amorphous fraction (RAF), and mobile amorphous fraction (MAF) [37]. Ordered crystalline chains constitute the CF phase, which is surrounded by macromolecular chains in the amorphous state [37,38]. Close to the CF lamellas, the neighboring polymer chains in amorphous state-denoted as tie molecules-undergo strong confinement effects and the reduction of segmental motions. As a consequence, this reduced molecular mobility of this confined amorphous phase may result in its own glass transition temperature, different from the glass transition of chains in MAF. The amorphous fraction close to the crystalline fraction is denoted as RAF. Identifying the location of ILs in the final LIonomers materials could help understanding the structuration mechanisms and their effect on the overall performance of LIonomers. The solid state Dynamic Mechanical Analyses reveal two relaxations of the amorphous, i.e., the ones of the MAF and RAF regions. Figure 7 and Table 2 indicate the temperature of the α and α’ relaxations related of the MAF and RAF fractions respectively, as well as the one of the β secondary relaxation (at −47 °C) of PPgMA. Such a relaxation corresponds to $T_{\beta }$ local segmental motions [39]. As various types of IL are combined with PPgMA, the temperature of this β relaxation supports the idea that ionic interactions don’t affect the local segmental motions of PPgMA. Similarly, $T_{\alpha }$ (MAF) of LIonomers associated to the glassy transition temperature remain the same as neat PPgMA one.

But the effect of P^+^DEP on the α’ relaxation (RAF) is more pronounced (93 °C of LIonomer L-P^+^DEP-10), which involves the mobility of the confined chains close to the crystalline lamellae [22,40,41]. The same trend could be also observed from L-P^+^DEP-2 but to a lesser extent since the ionic liquid content is lower. This effect may be attributed to the preferential localization of these ILs in the RAF phase. The chemical structure of ILs could explain their influence on the relaxation spectra of LIonomers. The strong interactions between ionic liquids and polar group within EMIM Acetate-based LIonomers highlighted by NMR and rheology account for the corresponding higher $T_{\alpha^{\prime }}$ value than that of EMIM DEP-based LIonomers. While for L-P^+^DEP-10, the alkyl chains of the phosphonium cation could promote the miscibility with PPgMA.

### 3.5. Effect of ILs on Crystallization of PPgMA

#### 3.5.1. Role of ILs in the Cystallization Process

Some works have already reported the influence of ILs on crystallization of semi-crystalline thermoplastics. For example, it was shown that phosphonium-based ILs could reduce the crystallinity ($X_{c}$) and crystallization temperature ($T_{c}$) of P(VDF-CTFE) [22,40]. In this work dedicated to maleic anhydride polypropylene/IL blends, no significant change of $X_{c}$ could be observed (see Table 3 from Appendix A). This phenomenon is related to the location and distribution of ILs within LIonomers. It agrees with the previous conclusions on the fact that PPgMA and ionic liquids exist as separated phases. The ion–ion and ion–dipole interactions occur at the interface between phases. On the other hand, the temperature of crystallization of PPgMA, $T_{c}$, increases with the addition of a small amount of P^+^DEP or EMIM DEP, i.e., ILs having a phosphate anion, and retrieve almost the value of neat PPgMA with 10 wt.% of IL. The addition of EMIM Ac has no influence on $T_{c}$ whatever its content.

DSC analyses of non-isothermal crystallization kinetics as well as observations by polarized optical microscopy (POM) can be used to detect the overall or partial crystallization process of PPgMA and LIonomers. The $\ln\left[-\ln\left(1-X_{t}\right)\right]$ vs. ln t plots according to Jeziorny’s non-isothermal crystallization model are shown in Appendix A. The value of $t_{1/2}$ (half-time crystallization) represents the time consumed to half crystallize, i.e., a longer time indicates a slower crystallization rate. As mentioned previously for the other parameters, $t_{1/2}$ slightly depends on (i) the nature of ion–ion and dipole–ion interactions within LIonomers that could delay the process of regular chain folding to generate the crystalline fraction, since the chain motions to change the configuration are restricted by nearby ILs involved in the interactions [22,40] ($t_{1/2}$: 0.86 min of L-EMIM DEP-10 vs. 0.77 min of PPgMA); (ii) the IL content, since high IL content may induce more interactions, followed by a greater impact on the crystallization process ($t_{1/2}$: 0.74 min of L-P^+^DEP-2 vs. 0.76 min of L-P^+^DEP-10; 0.72 min of L-EMIM DEP-2 vs. 0.86 min of L-EMIM DEP-10); (iii) the intensity of interactions could inhibit the movement of chain segment with some different extent, i.e., the restriction of chain motion caused by ion–ion interactions is stronger than that due to dipole–ion interactions. As a consequence, it results in a slower crystallization rate (L-EMIM DEP-10 vs. L-P^+^DEP-10); (iv) the miscibility of DEP anion in PPgMA promotes the crystallization process at low content.

In addition to the impact of the existing interactions within LIonomers on the crystallization process, IL itself may also influence the nucleation process. From the comparison of pure PPgMA matrix and LIonomers processed with 10 wt.% IL (at 5 K∙min^−1^, Table 3), L-EMIM Ac-10 crystallizes faster than all other systems. Because EMIM Ac could restrict the generation of nuclei (as shown in POM images a vs. d, a’ vs. d’ Figure 8) and cause a larger volume for the growth of the generated spherulites.

#### 3.5.2. Effect on Crystalline Phase

As shown in Figure 9, the PPgMA crystallizes according to the syndiotactic monoclinic system, mainly in α type crystalline form characterized by the Bragg diffraction peaks at 2θ ≈ 14.0° [110], 16.8° [040], 18.4° [130], 25.3° [060&150], 28.5° [220], and 42.6° [212]. A very weak diffraction peak belonging to (002) plane of the syndiotactic monoclinic [42] can also be detected at 23.6°. γ-form diffraction peak is not observed in the XRD profile (21.4° [α111&γ202]) because it is very difficult to distinguish γ-form from α-form diffraction peaks. γ-form cannot be present in this case, as the generation of this form requires specific crystallization conditions such as high pressure [43]. In LIonomers, most of diffraction peaks of PPgMA are not modified, but a new peak at 2θ ≈ 16.0° belonging to β (300) could be observed when 10 wt.% of EMIM DEP and EMIM Ac ILs are considered. The relative content of β-phase can be calculated from the Equation (7):(7)$$K=\frac{I_{300}}{I_{300}+I_{110}+I_{040}+I_{130}}$$
where $I_{300}$, $I_{110}$, $I_{040}$, and $I_{130}$ are the intensities of the corresponding monoclinic phase diffraction reflections at 2θ ≈ 16.0, 14.0, 16.8, and 18.4 respectively [44].

The generation of β-phase in LIonomers based on EMIM DEP and EMIM Ac is related to the existence of stronger ion–ion and ion–dipole interactions with PPgMA than ones generated with P^+^DEP. These interactions within L-EMIM DEP-10 and L-EMIM Ac-10 restrain the folding and rearrangement of polymers chains into ordered conformation, resulting in the β-phase which is less stable crystalline form than the α-form. On the other side, LIonomer based on 10 wt.% EMIM Ac exhibits larger β-form area than one obtained with the addition of 10 wt.% EMIM DEP (9.6% vs. 3.2%; see Table 4). As these two ILs hold the same cation (EMIM^+^), the extent-of-influence of these ILs on crystallinity morphology is associated with their interaction patterns (in NMR analysis) and the size of the anion (acetate vs. phosphate). More ion–ion positions contribute EMIM Ac to reinforce interactions, while the bigger and less interactive anion makes EMIM DEP less prone to promote strong interaction. The strongest ion–ion and ion–dipole interactions within LIonomers can be correlated to their rheological behavior.

### 3.6. Effect of ILs on Mechanical Behavior of LIonomers

Owing to the generated interactions and the structuration of the ILs within PPgMA matrix which can be tuned by the nature of ILs, i.e., the type of the anion and cation, the mechanical behavior of LIonomers can be also tailored. At low content of ILs, the Young’s modulus significantly increases whatever the nature of the ionic liquid, as shown in Figure 10 and Table 5. With the addition of a large amount of ILs, the modulus decreases significantly due to the presence of large separated IL-rich phases that act as defects except for L-EMIM Ac-10. In the latter case, strong ion–dipole interactions have been confirmed and the specific mechanical behavior of this LIonomer agrees with rheology results. On the other side, the strain at break increases dramatically for LIonomers based on imidazolium cation unlike phosphonium cation. Based on the DSC and WAXD results, especially crystallinity rates and morphisms, the generation of β-phase caused by strong ion–ion/dipole–ion interactions and multiscale morphologies, i.e., ILs as clusters and separated phase, can explain the difference of strain at break observed from the various types of LIonomers. The addition of EMIM Ac (even at low content, i.e., 2 wt.%) leads to the enhancement of both Young’s modulus and strain at break. The imidazolium cation plays a key role in the reinforcement of the mechanical properties. This cation leads to strong interactions with PPgMA that governs the macroscopic behavior of the corresponding LIonomers. The interactions based on phosphate anion are not sufficient to reinforce the mechanical behavior of the related LIonomers. The interactions are even stronger with the imidazolium cation and further when imidazolium is combined with acetate anion as more ion-ion interaction positions.

In summary, it is worth to noting that the introduced ionic liquids in grafted polypropylene don’t behave as plasticizers, i.e., leading to decrease the Young’s modulus and increase the strain at break as reported in the literature [22,40]. The mechanical behavior can be tailored to the nature of the cation and anion of the ionic liquid. For LIonomers, the slightly enhanced crystallinity rate contributes partly to the increase of stiffness while the multiscale distribution of ILs within the PPgMA matrix, i.e., as ionic multiplets and IL-separated phase, as well as strong ion–ion/dipole–ion interactions are the major contributors to the mechanical property of LIonomers.

## 4. Conclusions

From the combination of the maleic anhydride-grafted polypropylene (PPgMA) and three different ionic liquids, tributyl (ethyl)phosphonium diethyl phosphate (named P^+^DEP), 1-ethyl-3-methylimidazolium diethyl phosphate (named EMIM DEP), and 1-ethyl-3-methylimidazolium acetate (named EMIM Ac), LIonomers, i.e., a new generation of ionomers, have been successfully synthesized. This work reveals that the presence of grafted maleic anhydride on the polypropylene backbone as the “head group” combined with ionic liquid pairs could lead to the formation of new nanostructures via the generation of ionic and polar interactions. Different morphologies can be designed, i.e., from a phase separation resulting in a single-(nano) scale structuration composed of distribution of ionic clusters when LIonomers regarding phosphonium-based ILs, to a multi-scale structure combining ionic clusters but also IL-rich separated phase as for imidazolium cation-based ILs. The smaller size of the separated IL-rich phases is prone to establish the stronger ILs/polymer interactions. The imidazolium cation combined with the acetate anion appears as the most relevant IL to optimize the interactions with maleic anhydride group grafted onto the polypropylene backbone. The physical properties both in the molten and solid states appear to be strongly influenced by these interactions. In fact, the increase of the complex viscosity highlights the existence of IL/polymer interactions and the ability in such a case to form an ionic network. In relation with their morphology, crystallinity, and the existence of percolated ionic phase, LIonomers display both improved Young’s modulus and ability to sustain large strains before break. Therefore, these LIonomers are very promising for processing functional materials such as foams or melt blown membranes.

## Data Availability

Not applicable.

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
