# Peer review of "LIonomers-New Generation of Ionomer: Understanding of Their Interaction and Structuration as a Function of the Tunability of Cation and Anion"

_polymers, 2023, doi:10.3390/polym15020370_

Round 1
Reviewer 1 Report
This is a review for Hou et al. "Lionomers-New Generation of Ionomer: Understanding of Their Interaction and Structuration as a Function of the Tunability of Cation and Anion" submitted to "polymers".
Overall, the manuscript is missing a conclusive discussion of the "Understanding" and "Tunability" to do the words in the chosen title justice.
The following questions and comments should help improving the scientific quality of the manuscript.
In the introduction, the authors should provide more insight why out of the many ILs these three specific ILs were selected. The article could benefit from an explanation how the many analytical tools were planned to supplement each other for deriving the main conlusions.
What application of this class of materials do the authors envision?
Can we assume, these are safe materials to handle, process, and use?
What is the equilibrium water content under lab conditions? Is the water content in the different studies the same? How does water content influence the outcome of the respective studies? How does it affect the properties in possible applications?
What type of test bar (geometry) was used for tensile tests? Injection moulded parts? What were the conditions/parameters for injection moulding?
Was water content of the test parts controlled/tested for the tensile tests? How large was the sample size?
EDX results for element fractions should better not be given with 0.01 wt% precision, e.g. lines 145 ff. Or is the accuracy of EDX analysis really that good? Why is there no EDX from all mixtures/images?
The authors should refrain from using the word evidence as the data can also be interpreted in a different way.
The formatting of the Figures is not comprehensive and most of the text/labelling completely illegible.
Figure 2. I cannot read the tiny labels inside the pictures. What is the meaning of the black lines? Scale bars are illegible. The images should all be at the same scale to allow for a visual comparison.
In the TEM images, what causes the contrast between the two phases? How are the two phases attributed to the different domains in the material?
In NMR, does "dissolving" in chlorobenzene lead to dispersion or complete solution of the compounds? How does the solvent affect the IL-polymer chain interaction? This should enter the design of the experiments/analysis and the discussion of the results. Was solid-state NMR considered as an alternative that circumvents solution effects and better represents the bulk case?
Figure 3. Axis labelling illegible.
What is this symbol at the end of line 259? Is this an error?
Figure 8. The images were acquired at 120°C? I can hardly see a difference between any of the image pairs. I suggest the authors select different/further images and/or better explain/highlight what to look for.
In the conclusion, based on their findings, the authors should elaborate on how control of the resulting material properties by design of the blends can be accomplished. More of X leads to Y-behaviour. This manuscript would greatly benefit.
Line 413. Who wrote the manuscript?
Add Axis labelling in the Figures in the SI.
Element labels in S1 hard to read. Order of spectra unclear. "Spectre" is not the correct English term. The rest of the figure labelling should also be translated to English.
Author Response
Dear editor,
We would like to thank the reviewers for their careful reading and relevant comments that will improve undoubtedly the quality of the paper. All the remarks were been taken into account and corrections are brought in red in the paper.
Reviewer 1:
This is a review for Hou et al. "Lionomers-New Generation of Ionomer: Understanding of Their Interaction and Structuration as a Function of the Tunability of Cation and Anion" submitted to "polymers".
Overall, the manuscript is missing a conclusive discussion of the "Understanding" and "Tunability" to do the words in the chosen title justice.
The following questions and comments should help improving the scientific quality of the manuscript.
- In the introduction, the authors should provide more insight why out of the many ILs these three specific ILs were selected. The article could benefit from an explanation how the many analytical tools were planned to supplement each other for deriving the main conclusions.
Answer: The authors chose these three ILs by relying on the extended knowledge of the team that is working on more ten years on phosphonium and imidazolium based ionic liquids (ILs). ILs are selected to prepare LIonomers owing to their chemical nature. P+DEP exhibits a smaller steric hindrance than other phosphonium-based ionic liquids, leading to easier interactions with polar groups of polymer matrix. As written in the manuscript, the EMIM DEP is selected to change the nature of the cation while combining the same anion as P+DEP one in order to understand the role of cation within LIonomers. In addition, the nature of the anion is also considered by comparing the same cation, that’s why the EMIM Acetate is selected.
All the article is based on showing how the interactions between ILs and matrix take place by using NMR spectroscopy and how they pilot the structuration of polymers by using rheology and TEM microscopies and what it is the impact of interactions on mechanical properties. Some sentences in the text express that. For example: “such different interactions may connect the various dispersion of ILs in PPgMA and bring about the modifications into the overall performance (like molten viscosity, crystallization behavior, and so on)” in “PPgMA-ILs interactions”; “These cation-anion pairs could also self-assemble to form multiplets and lead to this double scale structuration observed by TEM” in “Effect of ILs on rheological properties: Strength of interactions”, and so on. In fact, the results from different tests can supplement each other, bringing proofs on the existence of prepared LIonomer-the new generation of ionomer firstly, then on the interactions within LIonomers and their arrangement tuned by the nature of ILs (by changing cation or anion).
- What application of this class of materials do the authors envision?
Answer: this kind of materials may be used for processing foams or melt blown membranes as written at the end of the manuscript. Nowadays, the ability of LIonomers to be used for processing foams as lightweight materials has been proved and will be published very soon.
- Can we assume, these are safe materials to handle, process, and use?
Answer: yes, these LIonomers are quite safe, and easy to handle during their processing in melt way and use. ILs are not toxic, the low contents of ILs ensure to be well embedded in the matrix and avoid the exudation. So LIonomers are quite safe.
- What is the equilibrium water content under lab conditions? Is the water content in the different studies the same? How does water content influence the outcome of the respective studies? How does it affect the properties in possible applications?
Answer: the processing temperature in melt way is 180 oC, thus the impact due to water can be ignored. In the same time the PPgMA is dried at 70oC for 12 hours to remove the water uptake even if it is minimal, since polypropylene is a hydrophobic polymer. All the prepared samples are stored in dry and cold environment, and placed in the oven at 80oC for 4 hours before test. Therefore, it can be assumed that all tests are not affected by water. When LIonomers are used for processing foams, high temperature and high pressure will be applied on samples without any influence due to water.
- What type of test bar (geometry) was used for tensile tests? Injection moulded parts? What were the conditions/parameters for injection moulding?
Answer: as written in the manuscript, the samples used for tensile test are obtained from DSM micro extruder, and they are dumbbell-shaped specimens at 30 oC.
- Was water content of the test parts controlled/tested for the tensile tests? How large was the sample size?
Answer: Once the dumbbell samples are prepared by injection, the tensile test is carried out. All the prepared samples are stored in dry and cold environment, and placed in the oven at 80oC for 4 hours before test. Therefore, there is no need for testing the water content.
The dimensions of dumbbell-shaped specimens are T=2mm, w0=12mm, w=4mm, L=30mm, D=45mm, L0=73.5mm, as shown
- EDX results for element fractions should better not be given with 0.01 wt% precision, e.g. lines 145 ff. Or is the accuracy of EDX analysis really that good? Why is there no EDX from all mixtures/images?
Answer: The EDX analysis is used to confirm the location of IL and to observe their dispersion state within polymer matrix. Therefore, it is not necessary to perform EDX on all samples. The values are given with a 10-2 precision.
- The authors should refrain from using the word evidence as the data can also be interpreted in a different way.
Answer: the word “evidence” has been replaced by other words, for example: show, indicate, demonstrate, contribute to, …
- The formatting of the Figures is not comprehensive and most of the text/labelling completely illegible.
Answer: all figures in the manuscript have been redone.
- Figure 2. I cannot read the tiny labels inside the pictures. What is the meaning of the black lines? Scale bars are illegible. The images should all be at the same scale to allow for a visual comparison.
Answer: all figures in the manuscript have been redone, and the scale bar is shown at the bottom left corner. The images in the manuscript have the same scale.
- In the TEM images, what causes the contrast between the two phases? How are the two phases attributed to the different domains in the material?
Answer: the contrast between two phases of polymer matrix and ionic enriched area can be attributed to the different intensities of light passing through, which is ultimately due to the different densities of the materials after the addition of ionic liquids. As discussed in the manuscript, the EDX results could be used to confirm ionic enriched region and the polymer matrix.
- In NMR, does "dissolving" in chlorobenzene lead to dispersion or complete solution of the compounds? How does the solvent affect the IL-polymer chain interaction? This should enter the design of the experiments/analysis and the discussion of the results. Was solid-state NMR considered as an alternative that circumvents solution effects and better represents the bulk case?
Answer: Polypropylene (as all polyolefin) is very difficult to completely dissolve in the solvent. Therefore, it is partially dissolved, and C6D5Cl is chosen as the NMR solvent because polypropylene has a relatively good solubility in it. Based on the structure of C6D5Cl, any interactions (for example H-bonding, etc.) cannot be formed between solvent and samples, therefore it cannot affect the structure or interactions within samples. Polypropylene cannot be broken into powder that is required by solid-state NMR to achieve good results. That’s why we used C6D5Cl as the solvent to solubilize PPgMA and detect the interactions within LIonomers.
- Figure 3. Axis labelling illegible.
Answer: this figure is redone, and clear right now.
- What is this symbol at the end of line 259? Is this an error?
Answer: that symbol is due to a typographical error. We corrected them and marked in red.
- Figure 8. The images were acquired at 120°C? I can hardly see a difference between any of the image pairs. I suggest the authors select different/further images and/or better explain/highlight what to look for.
Answer: yes, these images are observed at 120 oC, and used to observe the partial crystallization process of PPgMA and LIonomers. The comparisons of image PPgMA (a, a’) vs. L-P+DEP (b, b’) and PPgMA (a, a’) vs. L-EMIM DEP (c, c’) display that the P+DEP and EMIM DEP do not act as nucleation inhibitors, and the corresponding difference of crystallization speed could be explained by the interactions between PPgMA and ILs. However, with the addition of EMIM Ac from 2 wt.% to 10 wt.%, the crystallization process becomes more faster (obtained from reduced t1/2 value), this strange phenomenon can be fully explained by the fact that the EMIM Ac restricts the generation of nuclei, resulting in a larger volume of the generated spherolites, as discussed in the manuscript. That’s why the images d and d’ (L-EMIM Ac-10) exhibit less nuclei than that of images a and a’ (PPgMA).
- In the conclusion, based on their findings, the authors should elaborate on how control of the resulting material properties by design of the blends can be accomplished. More of X leads to Y-behaviour. This manuscript would greatly benefit.
Answer: yes, the manuscript has been corrected according to your suggestions, and marked with red.
- Line 413. Who wrote the manuscript?
Answer: the sentence is revised according to your suggestions, and marked with red.
- Add Axis labelling in the Figures in the SI.
Answer: all Axis labelling of figures in SI have been added.
- Element labels in S1 hard to read. Order of spectra unclear. "Spectre" is not the correct English term. The rest of the figure labelling should also be translated to English.
Answer: The detailed data are mentioned in the manuscript, (spectrum 1 of image c: C: 85.19wt%, N: 2.51wt%, O: 10.94wt%, P: 1.36wt%; spectrum 1 of image e: C: 96.89wt%, O: 3.11wt%; spectrum 1 of image f: C: 94.35wt%, O: 4.7wt%, P: 0.96wt%; spectrum 1 of image h: C: 90.11wt%, N: 1.92wt%, O: 7.97wt%). “Spectre” has been replaced by “Spectra”, and marked with red in the manuscript.
Reviewer 2 Report
A nice paper using ILs as additives to maleic anhydride-grafted polypropylene , generating so-called Lionomers. Th idea is sound, the experiments are well performed, but there are some weaknessess
line 55: IL names incomplete
line 196: ironicaly?
Fig 3 quality on the NMR spectra is relatively low. Especially those of the EMIM OAc. The peaks are very wide, compound looks not pure. The base line in the DEP spectrum is hardly there. Why that particular solvent chosen?
Please try to use the same color for the IL in all spectra and another one for the lionomers.
Although the NMR experiments show indeed a shift, especially in the aromatic protons, those do not seem enough to support the structures suggested as intermediates. Further, 3.g. 2D-NMR experiments are needed in order tu support those
Author Response
Dear editor,
We would like to thank the reviewers for their careful reading and relevant comments that will improve undoubtedly the quality of the paper. All the remarks were been taken into account and corrections are brought in red in the paper.
Reviewer 2:
A nice paper using ILs as additives to maleic anhydride-grafted polypropylene, generating so-called Lionomers. Th idea is sound, the experiments are well performed, but there are some weaknessess
- line 55: IL names incomplete
Answer: The incomplete names trihexyltetradecylphosphonium bis (trifluoromethylsulfonyl) and trihex-yltetradecylphosphonium bis 2,4,4-(trimethylpentyl) have been replaced by Trihexyltetradecylphosphonium bis (trifluoromethylsulfonyl) imide and trihex-yltetradecylphosphonium bis 2,4,4-(trimethylpentyl) phosphinate, respectively, as well as marked with red in the manuscript.
- line 196: ironicaly?
Answer: the word “ironicaly” is replaced by “ionically”, and marked with red in the manuscript.
- Fig 3 quality on the NMR spectra is relatively low. Especially those of the EMIM OAc. The peaks are very wide, compound looks not pure.
Answer: The NMR spectra of EMIM OAc should be good. Because all ILs are produced from TCI (https://www.tcichemicals.com/FR/fr?gclid=Cj0KCQiA-JacBhC0ARIsAIxybyMrX1cfEhB5WDdJAky0r9ttxeuuHllaZX3RXIcvDpLodPNrK46K5sEaAliPEALw_wcB), and the purity of EMIM OAc, EMIM DEP and P+DEP are higher than 94%, 96% and 96%, respectively.
- The base line in the DEP spectrum is hardly there. Why that particular solvent chosen?
Answer: the NMR spectrum of DEP is clearly modified.
The testing temperature is 100 oC, C6D5Cl was chosen as the solvent because of its high boiling point (130 oC) and it is one of solvent able to solubilize at high temperature.
- Please try to use the same color for the IL in all spectra and another one for the lionomers.
Answer: according to your suggestions, all spectra curves of IL is using blue, while another colour (maroon) is applied for all LIonomers.
- Although the NMR experiments show indeed a shift, especially in the aromatic protons, those do not seem enough to support the structures suggested as intermediates. Further, 3.g. 2D-NMR experiments are needed in order to support those
Answer: according to the possible reactions between the ILs (cation/anion pairs) and MA groups of PPgMA, the intermediates and final LIonomers structures are proposed in the article. These intermediates are converted into the final products completely, since the molar ratio of IL/MA is higher than 1 when adding 10 wt.% IL. Meanwhile, all reactions between ILs and PPgMA are completed within five minutes under 180 oC. As a consequence, the direct proof of the intermediated structure is very difficult to obtain. But these intermediates are reasonably inferred from the structure of raw materials and products, and this inference is tenable in the context of the whole text. Because the NMR results can initially infer the reaction between ILs and MA as well as the final LIonomers produced, which can be corroborated with the conclusions of other properties, such as rheology (the improved viscosity at high temperature), structuration (especially the reinforced glass transition temperature), crystallization and mechanical properties in the manuscript. Therefore, it is not necessary to do 2D or 3D NMR testing.
Author Response
Dear editor,
We would like to thank the reviewers for their careful reading and relevant comments that will improve undoubtedly the quality of the paper. All the remarks were been taken into account and corrections are brought in red in the paper.
Reviewer 3:
This article is well documented and discussed and should be accepted, after minor revisions.
- Hou et all synthesized Lionomers, that are new materials based on the combination of IL and polymers. They synthesized three different ones, when they study the effect of cation and anion of IL with maleic anhydride-grafted polypropylene (PPgMA). They studied phosphonium and imidazolium cation and acetate and diethyl phosphate anion. They conclude and demonstrate imidazolium cation combined with the acetate anion appears as the most relevant IL to optimize the interactions with maleic anhydride group grafted onto the polypropylene backbone.
1: In Scheme 1, when authors design the structure of three different ILs, the first IL, tributyl (ethyl) phosphonium diethyl phosphate (P+DEP), do not have anion well designed.
Answer: According to the reviewer’s advice, we have best designed the anion structure.
Round 2
Reviewer 1 Report
Thank you for the detailed answers. I think, the manuscript has improved much. I still have a number of remarks - especially the figures did not improve adequately.
Small comment: P9, L263: not a "fact" --> an "idea", "theory" or "proposition"
Further comments to the answers to Reviewer 1:
Answer to 1: The manuscript really benefits from the improved introduction. In my opinion, this explanation is still missing in the manuscript: >>ILs are selected to prepare LIonomers owing to their chemical nature. P+DEP exhibits a smaller steric hindrance than other phosphonium-based ionic liquids, leading to easier interactions with polar groups of polymer matrix. As written in the manuscript, the EMIM DEP is selected to change the nature of the cation while combining the same anion as P+DEP one in order to understand the role of cation within LIonomers. In addition, the nature of the anion is also considered by comparing the same cation, that’s why the EMIM Acetate is selected.<<
Answer to 4: please add your statements about water content to the manuscript. But keep in mind, you graft maleic anhydride and add Ionic Liquid so the compound might not take up more water than you think. And the final properties of the materials in a possible product at application conditions would be affected by water content (like Young's modulus etc.). Did you find water in the NMR?
Answer to 5 & 6: please add the information about the test part geometry to the manuscript.
Answer to 7: The EDX results for element fractions should not be given with 0.01 wt% precision. You should show at least two EDX spectra from the same TEM image to convince the reader of the correct attribution of the light and dark phases. In Figure 2 h it is impossible to see what is in the image where "spectrum 2" was acquired.
Answer to 9: really not adequately redone: still unable to read labels and scale bars in Figures 2,3,4 8 and Figures 2 and S1 still not all in English
Answer to 12: Thank you for the explanation, please put this information in the manuscript. First, however, decide whether PP "is very difficult to dissolve" or "has a relatively good solubility" in C6D5Cl: >>Polypropylene (as all polyolefin) is very difficult to completely dissolve in the solvent. Therefore, it is partially dissolved, and C6D5Cl is chosen as the NMR solvent because polypropylene has a relatively good solubility in it. Based on the structure of C6D5Cl, any interactions (for example H-bonding, etc.) cannot be formed between solvent and samples, therefore it cannot affect the structure or interactions within samples.<<
Answer to 15: please put this info in the manuscript: >>images were acquired at 120°C<< it is not clear from the text/caption
Answer to 19: Why is the order e, f, h and then c? Why is there no spectrum 1 in h? (Or where is it?)
Author Response
Dear editor,
We would like to thank the reviewers for their careful reading and relevant comments that will improve undoubtedly the quality of the paper. All the remarks were been taken into account and corrections are brought in red in the paper.
Reviewer 1:
Thank you for the detailed answers. I think, the manuscript has improved much. I still have a number of remarks - especially the figures did not improve adequately.
1. Small comment: P9, L263: not a "fact" --> an "idea", "theory" or "proposition"
Answer: according to your suggestion, the word “fact” has been changed into “idea”.
Further comments to the answers to Reviewer 1:
2. Answer to 1: The manuscript really benefits from the improved introduction. In my opinion, this explanation is still missing in the manuscript: >>ILs are selected to prepare LIonomers owing to their chemical nature. P+DEP exhibits a smaller steric hindrance than other phosphonium-based ionic liquids, leading to easier interactions with polar groups of polymer matrix. As written in the manuscript, the EMIM DEP is selected to change the nature of the cation while combining the same anion as P+DEP one in order to understand the role of cation within LIonomers. In addition, the nature of the anion is also considered by comparing the same cation, that’s why the EMIM Acetate is selected.<<
Answer: according to your suggestions, these sentences have been mixed in the manuscript and marked with red (see P2, L69-L75).
3. Answer to 4: please add your statements about water content to the manuscript. But keep in mind, you graft maleic anhydride and add Ionic Liquid so the compound might not take up more water than you think. And the final properties of the materials in a possible product at application conditions would be affected by water content (like Young's modulus etc.). Did you find water in the NMR?
Answer: yes, according to your suggestions, the statement to avoid the influence of water has been added in the manuscript, and marked with red (see L144-L146, P4).
Yes, the final modulus could be affected by water, that’s why every prepared sample was stored in the cold and dry conditions. The measures to get rid of the effect of water had been carried out, therefore, without any water was obtained in NMR.
Answer to 5 & 6: please add the information about the test part geometry to the manuscript.
Answer: according to your suggestions, the geometry of dumbbell-shaped specimens has been added into the manuscript, and marked with red (see P4, L142-L144).
Answer to 7: The EDX results for element fractions should not be given with 0.01 wt% precision. You should show at least two EDX spectra from the same TEM image to convince the reader of the correct attribution of the light and dark phases. In Figure 2 h it is impossible to see what is in the image where "spectrum 2" was acquired.
Answer: the values in element results in 0.01 wt% is for more precise expression, but according to your suggestions, they are modified with 1wt%.
More element images are added in the Figure S1, and they are ordered according to the spectra number. In addition, the figure 2h is changed into the Figure S1, and more EDX images are introduced there.
Answer to 9: really not adequately redone: still unable to read labels and scale bars in Figures 2,3,4 8 and Figures 2 and S1 still not all in English
Answer: all figures have been redone according to your suggestions. The scale bars of Figures 2 and 8 locate at the bottom left corner, in white and red colour respectively. At the same time, their length is highlighted in corresponding caption at L170, P5 and L328, 12, and marked with red. In addition, Figures 3 and 4 are very clear now.
Answer to 12: Thank you for the explanation, please put this information in the manuscript. First, however, decide whether PP "is very difficult to dissolve" or "has a relatively good solubility" in C6D5Cl: >>Polypropylene (as all polyolefin) is very difficult to completely dissolve in the solvent. Therefore, it is partially dissolved, and C6D5Cl is chosen as the NMR solvent because polypropylene has a relatively good solubility in it. Based on the structure of C6D5Cl, any interactions (for example H-bonding, etc.) cannot be formed between solvent and samples, therefore it cannot affect the structure or interactions within samples.<<
Answer: according to your suggestions, these sentences have mixed in the manuscript and marked with red (see L100-L102, P3).
Answer to 15: please put this info in the manuscript: >>images were acquired at 120°C<< it is not clear from the text/caption
Answer: yes, you can read it at the end of L327, P12.
Answer to 19: Why is the order e, f, h and then c? Why is there no spectrum 1 in h? (Or where is it?)
Answer: such order is used to easy to compare and discuss in the manuscript. I have redone these images and put them in the manuscript. As for the number of spectrum in EDX, I have reordered them. In fact, by detecting the content of every element, we would like to confirm the location of added ionic liquids as well as the polymer matrix within LIonomers.
